# Mechanical dissipation from charge and spin transitions in oxygen-deficient SrTiO$_3$ surfaces

Marcin Kisiel[1], Oleg O. Brovko[2], Dilek Yildiz [1], Rémy Pawlak [1], Urs Gysin[1], Erio Tosatti[2,3] & Ernst Meyer[1]

Bodies in relative motion separated by a gap of a few nanometers can experience a tiny friction force. This non-contact dissipation can have various origins and can be successfully measured by a sensitive pendulum atomic force microscope tip oscillating laterally above the surface. Here, we report on the observation of dissipation peaks at selected voltage-dependent tip-surface distances for oxygen-deficient strontium titanate (SrTiO$_3$) surface at low temperatures ($T = 5$ K). The observed dissipation peaks are attributed to tip-induced charge and spin state transitions in quantum-dot-like entities formed by single oxygen vacancies (and clusters thereof, possibly through a collective mechanism) at the SrTiO$_3$ surface, which in view of technological and fundamental research relevance of the material opens important avenues for further studies and applications.

[1] Department of Physics, University of Basel, Klingelbergstrasse 82, 4056 Basel, Switzerland. [2] The Abdus Salam International Centre for Theoretical Physics (ICTP), Strada Costiera 11, 34151 Trieste, Italy. [3] Scuola Internazionale Superiore di Studi Avanzati (SISSA), and CNR-IOM Democritos, Via Bonomea 265, 34136 Trieste, Italy. Correspondence and requests for materials should be addressed to M.K. (email: marcin.kisiel@unibas.ch) or to E.M. (email: ernst.meyer@unibas.ch)

Strontium titanate (SrTiO$_3$ or STO for short) stands out among other oxides as a material with broad spectrum of physical phenomena and of functional properties. Throughout the last decades the interest in STO has increased due to its popularity as a versatile substrate for oxide electronics research and engineering[1,2]. Extreme electron mobility and superconducting properties[3–5], quantum paraelectricity in bulk[6], itinerant, impurity, and vacancy-based magnetism[2,7,8] of STO have been subjects of fundamental studies. In particular, charge trapping by oxygen vacancies ($V_O$)[9] and vacancy-related magnetism[10–13] are pertinent to the present study. Pristine STO develops oxygen vacancies when grown or annealed under oxygen-poor conditions[14], bombarded with noble gas ions[15], or under intense laser or ultraviolet irradiation[16]. When in numbers, $V_O$ can lead to the formation of two-dimensional electron gas (2DEG) on the surface of bare STO[17–20]. Moreover, oxygen vacancies were shown to be inherently magnetic both in bulk STO[12,21,22] and at its surface[5,13,23–29]. They were shown to exhibit either local uncorrelated magnetic moments[21,30] or stable long-range magnetic order, when present in sufficient concentration[5,12,26,29,31–33]. Although the influence of bulk and surface oxygen vacancies on the electronic and optical properties of STO is long known[34,35], the options for controlling these vacancies are poorly explored and offer a lucrative field for further research.

Atomic force microscopy (AFM) cantilevers, oscillating like tiny pendula over a surface[36], are primarily designed to measure extremely delicate non-contact form of frictional dissipation and serve as an ultra-sensitive, non-invasive spectroscopy method[37] (see Supplementary Methods Fig. 1 for details). Apart from provoking such conventional forms of non-contact energy transfer as phonon and Joule ohmic dissipation[36,38], the external perturbation caused by an oscillating tip might push a finite quantum system, or a collection of them towards a transition or a level crossing with subsequent relaxation of the system. A level crossing implies a dissipation channel for the external agent provoking the change[37,39]. For AFM this leads to distance and bias-voltage-dependent dissipation of the tip oscillation energy. Such dissipation was observed at the sudden injection of phase slips into a charge density wave on NbSe$_2$ surface[40] and at the crossing of electron energy levels in mesoscopic single-electron InAs quantum dots[39].

Here we measure the dissipation experienced by a sharp AFM tip oscillating at a range of lateral amplitudes (100 pm to 5 nm) over an oxygen-deficient SrTiO$_3$(100) single crystal surface cooled down to low temperatures of $T = 5$ K. At selected tip-surface distances and bias voltages, dissipation peaks are observed in the cantilever oscillation. We attribute these peaks to charge and spin state transitions in individual or possibly collective groups of oxygen vacancies, which act as natural quantum dots at oxygen-deficient surfaces of STO, as suggested by a recent first principles calculations[13]. We further discuss possible pathways for electrons, which could give rise to these transitions under the action of the time-periodic electrostatic or van der Waals potential exerted by the AFM tip. The dissipation peaks appear at large crystal reduction (therefore with many surface vacancies), and at long tip-sample distances (even above 10 nm), disappearing above $T = 80$–90 K. While these elements point to a collective mechanism involving many vacancies, we find that the crude single vacancy model already provides a very helpful level of understanding.

## Results

**Sample characterization with STM.** Surface morphology and chemical composition of STO are known to change significantly under high-temperature annealing[41]. We thus start our study by examining the topography of STO with a scanning tunneling microscope (STM) between cycles of annealing at subsequently increasing temperatures (see "Methods" for the measurement protocol and Supplementary Figure 2 for more temperature-dependent measurements). While rough and irregularly-shaped at low annealing temperatures, STO surfaces develop a well-defined terrace and island structure after prolonged annealing at about 800 °C forming a range of reconstruction patterns [(2 × 2), (2 × 1) and dominant c(4 × 2)] yet retaining a relatively high surface roughness[41,42]. At about 900 °C well ordered step-terraces develop and acquire smaller reconstruction unit cells of (2 × 2) and arising from oxygen vacancies ($\sqrt{5} \times \sqrt{5} - R26.6°$), a behaviour which persists at higher annealing temperatures[42]. Representative constant current topographies after annealing at $T = 850$ °C and 1050 °C are shown in Fig. 1a–c, respectively, exhibiting, as mentioned, well formed terraces and roughness at the atomic scale after annealing to $T = 850$ °C and atomically flat terraces with clear ($\sqrt{5} \times \sqrt{5}$) reconstruction after 1050 °C[42]. The atomic resolution STM image [Fig. 1c] of the 1050 °C-annealed surface with $V_{bias} = 1$ eV shows several tetragonal reconstruction domains of three different orientations[43]. It exhibits a variety of structural peculiarities, which appear as either points or extended areas/lines of increased or decreased apparent height. Bright features can be attributed to adatoms on top of the reconstructed ad-layer or oxygen hydroxyl (O−H) groups resulting from rest gases in the vacuum chamber, as it is known[44] that interstitial Ti atoms can cause a substantial increase in local surface reactivity and thus facilitate O−H group adsorption. Dark features could, on the other hand, indicate missing ad-islands in the ($\sqrt{5} \times \sqrt{5}$) reconstruction[11,45] or, considering that annealing was carried out

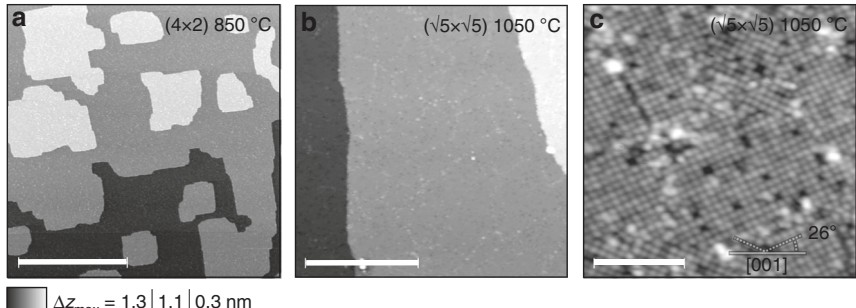

**Fig. 1** STM topography of the STO surface. Overview of the c(4 × 2) reconstructed SrTiO$_3$ (100) surface after long-term annealing at $T = 850$ °C (**a**) and ($\sqrt{5} \times \sqrt{5} - R26, 6°$) reconstructed SrTiO$_3$ (100) obtained after annealing to $T = 1050$ °C ((**b**, **c**))[41,42]. Images are constant current topographies with tunneling parameters $U_{tip} = 2$ V, $I = 30$ pA (**a**) and $U_{tip} = -1$ V, $I = 10$ pA (**b**, **c**). Dark features visible after high-temperature annealing (**c**) are related to oxygen vacancies[42,46]. The length of the scale bar is 200 nm, 50 nm and 10 nm for **a**–**c** images, respectively

at oxygen-poor conditions, more likely represent oxygen vacancies ($V_O$) at or near the surface[42,46]. In agreement with that, the density of empty states 1 eV above Fermi level, which the STM probes, is high at all sites where a surface O atom is present.

**Non-contact friction measurements**. After the topographic characterization of the surface between the annealing cycles, we measure the non-contact frictional dissipation coefficient $\Gamma$ as a function of the applied tip-sample bias voltage. Measurements are taken with a stiff gold-coated tip oscillating in pendulum geometry at a moderate planar amplitude of 1 nm while kept at a constant tip-sample distance of 15 nm (see "Methods" section for details and the definition of $\Gamma$ and Supplementary Fig. 1 for a sketch of the setup). Dissipation spectra for samples annealed at $T = 850\,°C$ and $950\,°C$ (see also Supplementary Fig. 3 for morphology characterization) are shown in Fig. 2. Both spectra were taken on flat terraces, far from the step edges. The sample annealed at lower temperature reveals a spectrum, which displays a parabolic curve characteristic of Joule non-contact friction[38]. Similarity with Nb metal above $T_C$[36] indicates some ohmic electron conduction in the STO surface. The dissipation spectrum of the $950\,°C$ annealed sample, however, exhibits in addition two distinct dissipation peaks at $V = V_{CP} \pm 3.25\,V$ located symmetrically around the contact potential voltage $V_{CP} = -0.7\,V$ and a broader, less pronounced dissipation peak around 4.2 V. Moreover, dissipation spectra obtained with a local probe (metallic tip tuning fork sensor) systematically shown an identical, single dissipation peak for $(\sqrt{5} \times \sqrt{5})$ reconstructed STO after annealing at $T = 1050\,°C$ (see Supplementary Fig. 4). It is the origin of those dissipation peaks, and the physical information on the nanotechnologically important surface oxygen vacancies that the rest of this paper shall be devoted to.

First, we use the scanning tunneling spectroscopy (STS) mode of our AFM to measure at closer tunneling tip-surface distances the $dI/dV$ spectra (Fig. 3) above the $(\sqrt{5} \times \sqrt{5})$ reconstructed areas and above the apparent dark spots in Fig. 1c. The main difference between the spectra of reduced and non-reduced STO is an additional defect state in the gap ($O_A$ in Fig. 3) lying at about 1.2 eV below the Fermi level, absent for samples annealed at moderate temperatures (for details about the defect state position see Supplementary Fig. 5). Published data[42] attribute a similar STS peak to reduced STO and surface oxygen vacancies, suggesting that the dark STM spots mark vacancies formed in the process of high-temperature annealing. Another feature of the differential conductance spectra of $V_O$ is a weaker peak at about 0.3 eV below the Fermi level ($O_B$ in Fig. 3). It can be attributed to a shallower in-gap state of the $V_O$, which, as shown in our recent theoretical study[13] can be gradually emptied by changing the local chemical potential of the surface under the scanning probe tip (see Supplementary Fig. 6 for details).

It is therefore natural to assume that the non-contact dissipation peaks (Fig. 2) measured with a AFM tip swinging near the vacancy site should be due to electronic transitions within the vacancy or between the vacancies under the influence of the AFM tip's potential field changes. In this experiment the large tip-surface distance and its large swing implies that in general more vacancies are likely to be collectively involved in each dissipation event and peak, in a manner that is hard to resolve. The interpretation is greatly helped, we find, by a single vacancy model, as follows. An isolated $V_O$ at the surface of STO can be regarded as a quantum dot with a discrete set of charge states, a dissipation peak would arise in the tip oscillation from a periodic transition between them. This conclusion may also entrain particularly exciting consequences, as the charge state of an $V_O$ at an STO surface is uniquely linked to its magnetic state[13]. The dangling bonds of the Ti atoms bordering $V_O$ (see Supplementary Fig. 6 for a sketch) are known to host the two

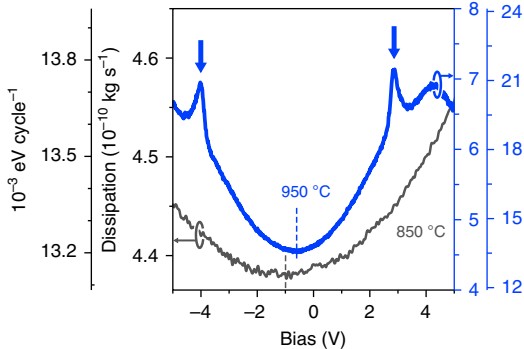

**Fig. 2** Dissipation peak formation upon high-temperature annealing of reduced STO. Bias voltage dependence of friction coefficient $\Gamma$ between the metallic tip oscillating with amplitude $A = 1\,nm$ and SrTiO$_3$ (100) sample annealed at two different temperatures of $850\,°C$ (black) and $950\,°C$ (blue). Higher annealing temperatures (characterised by $(\sqrt{5} \times \sqrt{5})$ reconstruction) lead to larger dissipation and pronounced dissipation peaks marked by arrows. Additional grey axis on the left and pale-blue axis on the right give corresponding per-cycle energy dissipation values $P[eV/cycle]$ (see "Method" section for details)

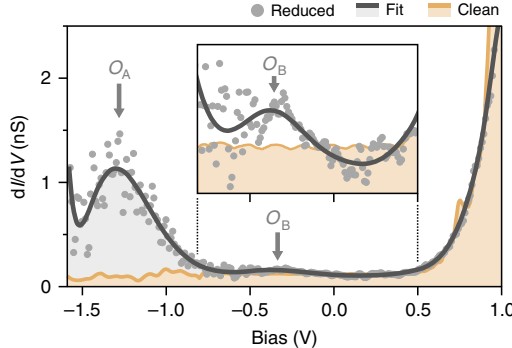

**Fig. 3** Electronic characterisation of oxygen vacancies at SrTiO$_3$ surface. The $dI/dV$ spectrum of the reduced surface is characterised by the presence of a broad peak localized at about 1.2 eV below the Fermi level ($O_A$), a signature of an oxygen-deficient system[42]. A smaller peak $O_B$ just below the Fermi level (see inset) can also be attributed to oxygen impurity levels (see "Discussion"). For comparison, the spectrum for non-reduced sample is shown in beige (light brown)

electrons, or fractions thereof, left behind by the departed O atom. Calculations show that these electrons occupy dangling bond $d$-localized states corresponding to levels within the band gap of STO[13]. Depending on the local value of the electron chemical potential the system can be coerced into an $S = 0$ charge-neutral $q = 0$ singlet resulting from antiferromagnetically coupled Ti with oppositely aligned 1/2 spins, the 1/2 doublet state of a $q = -1\bar{e}$ charged single-hole vacancy, and the magnetically dead two-hole ($q = -2\bar{e}$) vacancy state[13]. The local chemical potential in our experiment is controlled by the AFM tip voltage and tip-sample separation. The tip oscillation may thus cause time-periodic transitions of the quantum dot between its different states. A distant tip acts as a pure potential source without any tunneling current, while the vacancy state transition involves a change of charge besides spin. It is necessary, therefore, to understand the physical mechanism by which the charge and spin state transition may be provoked here (see Supplementary Fig. 6).

**Local dissipation spectra taken with a stiff cantilever tip**. Key clues that clarify the tip-induced $V_O$ transition mechanism are

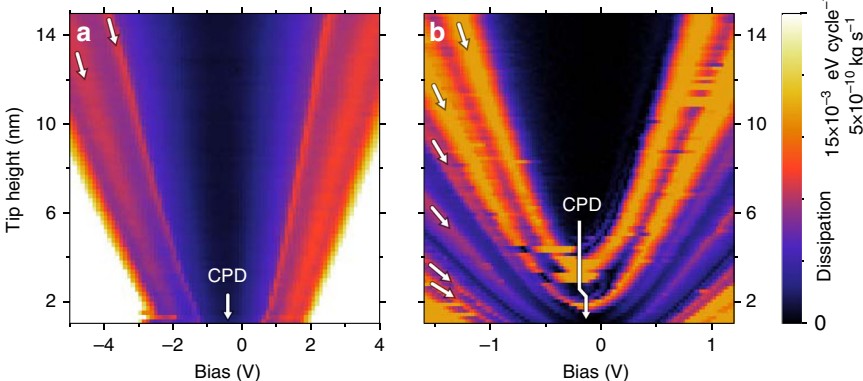

**Fig. 4** Local and non-local bias voltage $V$ and tip-sample distance $d$ resolved dissipation maps $\Gamma(V,d)$. The dissipation maps were taken with two different AFM tips suspended in pendulum geometry: **a** the stiff gold-coated cantilever tip with a moderately low stiffness of 50 N/m driven at a moderate nominal amplitude of 1 nm and **b** an ultra-sensitive probe with very low stiffness of (0.1 N/m) driven at oscillation amplitudes of up to 5 nm. See also Supplementary Fig. 7 for more details about dissipation map

provided by the dependence of the dissipation behavior on the tip-sample distance $z$. For distances spanning a range from near-contact regime to 15 nm, we measure the bias ($V$) dependence of the AFM frictional dissipation coefficient $\Gamma$, thus obtaining a full $\Gamma(V,d)$ map. We use two different tips in pendulum geometry: (i) a stiffer gold-coated cantilever tip with a moderately low stiffness of 50 N/m operated at a nominal amplitude of $A = 1$ nm and (ii) an ultra-sensitive probe with very low stiffness of (0.1 N/m), which implies larger oscillation amplitudes of up to 5 nm. Unlike the stiff cantilever the soft sensor is not gold-coated, which results in a very high value of $Q$ but requires significant doping of the silicon tips for attaining reasonable conducting properties.

The stiff tip dissipation map taken above a reconstructed section of the STO surface is shown in Fig. 4a. It shows features found in other similar experiments[36]. For each tip-sample distance the friction coefficient exhibits a parabolic behavior (see gray curve in Fig. 2, which is a cross-section of the map in Fig. 4a at $d = 15$ nm), hallmark of Joule-loss non-contact friction[36,38]. The map is near-symmetric with respect to the contact potential difference (CPD) voltage marked in the plot. The outer hull of the dissipation map is defined by a the critical bias voltage for excitation of electrons over the band gap of the STO, signaling a sharp increase in dissipation (about $\pm 2.5$ V) at close range. The sharp dissipation peaks at the sub-bandgap voltages marked with arrows both reach $\Gamma = 5 \cdot 10^{-10}$ kg/s, maintaining roughly the same intensity independently of the tip-sample distance. The peaks are observed at non-zero (with respect to CPD) biases even at close range and shift towards higher bias voltages with increasing tip-sample distance, which indicates that the effect is voltage rather than force controlled, similar to the case of quantum dots[39] and in analogy to quantum dots the amount of dissipated energy given by different sensors is also in the order of 10–20 meV/cycle.

**Dissipation spectra taken with a soft cantilever tip**. To further investigate the dissipation mechanism, we repeat the dissipation measurements with an ultra-sensitive doped-silicon sensor. Very low stiffness ($k = 0.1$ N/m) of the sensor implies larger horizontal oscillation amplitudes of up to 5 nm and the absence of metal coating might imply different interaction behavior. The non-contact friction dependence map $\Gamma(V,d)$ taken with a soft tip is shown in Fig. 4b. Here again bright features correspond to the high dissipation maxima up to $\Gamma = 2 \cdot 10^{-10}$ kg/s. However, instead of just two dissipation peaks the soft non-local sensor sports a whole family of dissipation maxima. We noticed a substantial change of CPD value ($\Delta$CPD = 0.1 V) indicated by the

elbow shape of the arrow in Fig. 4b in the transition while switching from one to the next dissipation maxima trace. It is consistent with the idea of the oscillating tip causing a persistent accumulation of charge below it and further corroborates the guess that the charging dissipation scenario is responsible for the dissipation peaks. These again proves electronic nature of energy dissipation in analogy to refs. [39,47].

## Discussion

Charge transfer mechanisms: The dissipation peak phenomenology fits a model of charge and spin state transitions in oxygen vacancies present at reduced STO surfaces, as follows. The distant pendulum tip projects a potential shadow on the underlying $SrTiO_3$ (001) surface deforming the local electronic chemical potential (either directly or indirectly by distorting the surface atom positions). This perturbation is larger for a charged tip, but even at zero voltage, the van der Waals polarization interaction shifts the chemical potential by modifying the electron self-energy. If one or more oxygen vacancies happen to lie in the potential shadow of the tip (see Fig. 5a for a sketch) the vacancies closer to the tip's position experience a different electron chemical potential than those at the periphery. At certain values of the imposed potential (the strength/depth of which is dependent on the tip-sample distance and mutual bias) the topmost filled impurity level of one of the $V_O$ can become aligned with an empty electron reservoir, thus provoking one electron to leave the vacancy, causing a transition[13]. Some of the possible transition paths are sketched in Fig. 5b. Electrons can be exchanged between a neutral vacancy and a charged one, or a vacancy and the conduction band of the surrounding STO surface as well as the 2DEG known to exist on the doped interfaces of STO[4,19]. Oxygen vacancies on STO surfaces are known to exist as both isolated entities or as cluster defects. The latter can exhibit multiple impurity levels and reside in a number of charge states[13]. Alignment of one or several impurity levels of such an extended vacancy with either a single-vacancy impurity level or the STO electron bath can lead to one or several consecutive electron transitions, taking place by resonant tunneling or generally by electron transfer. An oscillating AFM tip at a fixed bias can cause, when it reaches the right distance, a sufficient deformation of the local chemical potential to provoke one of these transitions, which is reversed as the tip oscillates back. Every single electron transfer channel leading to a transition can manifest itself as AFM tip dissipation peaking precisely when the bias voltage and distance first reach the values required to activate the transfer. The

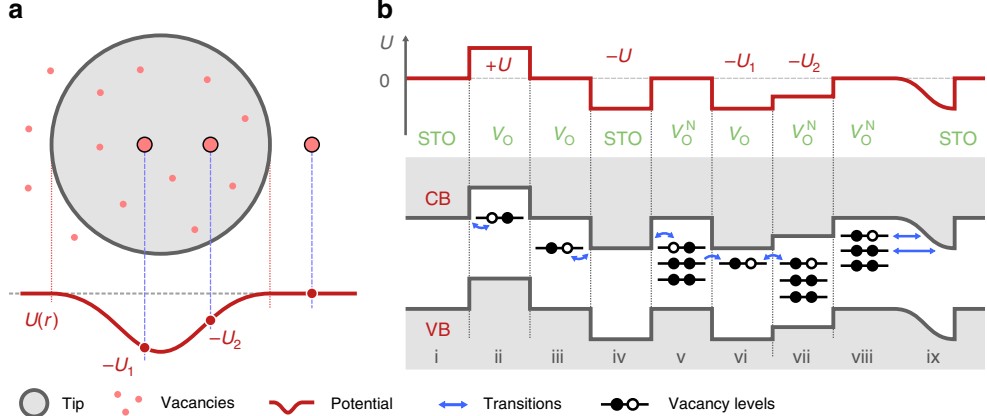

**Fig. 5** Electronic transitions in oxygen vacancies caused by the potential of an oscillating AFM tip. **a** AFM tip, through electrostatic and van der Waals interactions, modifies the electron local potential at the surface in an area with one or several oxygen vacancies, altering their chemical potential (Fermi level) with respect to each other and the surrounding STO. Local alteration of chemical potential and electron level shifting can lead to a realignment of the impurity levels of $V_O$ facilitating electron transfer and provoking charge state transitions in the system. Several possible paths thereof are sketched in **b** where scenario (ii) and (iii) correspond to electron exchange between single vacancy and conduction band of SrTiO$_3$, (v), (vi), (vii)—electron exchange between group of vacancies, and (viii), (ix)—electron exchange between vacancies and surface 2DEG

approximately symmetric pattern for positive and negative tip voltage suggests that this mechanism is reversible.

In the dissipation measurements carried out with the stiffer gold-coated tip, the area swept is much smaller, and the two observed dissipation peaks can originate from the single and double discharging of a standalone vacancy with electrons being transferred to either other vacancies/vacancy clusters or to the STO conduction band. We estimate the average distance between the dark defects on our sample from STM images to be 3.7 ± 1.6 nm (see Supplementary Fig. 8). Thus a stiff tip swinging with an amplitude of 1 nm is unlikely to impact more than one vacancy, even considering that the characteristic dimensions of the perturbation caused by the tip suspended at several nm distance shall be larger than the amplitude of its swing. The softer non-local tip swinging at an amplitude of 5 nm will, on the contrary, enclose in its potential shadow path several $V_O$ or an $V_O$ cluster resulting in more electron transition channels and consequently more dissipation peaks. The fact that in the latter case some dissipation peaks exist even in the case when the CPD is completely compensated is a likely residual effect of the van der Waals interaction, as the $V_O$ impurity level is a relatively shallow one and does not require much perturbation to be emptied. This assumption is especially important to us since $V_O$ charge states were shown to be stable with respect to mechanical perturbations of the system[13] thus making the influence of the tip on the observed charge and spin-state transition much more likely to be electronic than mechanical in origin. Unfortunately, we cannot specify exactly what levels we are precisely talking about. Comparison of our STS data with DFT calculations (see Supplementary Figs. 6, 9; also ref. [13]) suggests that for a single vacancy charging scenario, the observed dissipation might arise from the transitions between $q = -2e$ and $q = -1e$ charge states. Interestingly, that should also involve a spin transition, from $S = 1/2$ of the $q = -1e$ state, to $S = 0$ in the $q = -2e$ state, where the two electrons in the two Ti oppositely facing dangling bonds couple antiferromagnetically[13]. Finally the temperature-dependent measurements (see Supplementary Fig. 2) reveal that the dissipation peaks, which we attribute to charging and discharging of Oxygen vacancies persist up to a temperature of 80–90 K. At those temperatures the charging energy is much larger than thermal energy $\left(\frac{e^2}{2C} \gg k_B T = 7 - 8\,\text{meV}\right)$ and the occupancy of the $V_O$ can be controlled with single-electron precision, hinting at a wide range of promising functionalities, and suggesting that the "shadow

potential" projected by the tip works within an energy scale of order $k_B T$. This is not unreasonable, since the vacancy redox levels, all lie very close to the SrTiO$_3$ conduction band and can possibly be switched by a 8 meV tip-induced shift.

We mention, before closing, a work that reported a single noncontact AFM frictional peak on STO and on NbSe$_2$[48], as well as theoretical modeling that rationalised it in terms of a defect spin model[49]. The NbSe$_2$ data show that the peak maximum coincides with the earliest onset of weak tip-surface electron tunneling, and the simultaneous onset of stiffening of the tip frequency. All our AFM dissipation peaks occur at large distances where tunneling is totally absent and where the tip frequency is softened rather than stiffened (see Supplementary Fig. 4). Both elements indicate a dissipation origin different from that of Ref.[48].

To summarize, our sensitive low-temperature ($T = 5$ K) AFM experiments with soft cantilever tips show single or multiple mechanical dissipation peaks over a oxygen reduced surface of STO. We attribute them to the tip coupling to charge and spin state transitions in quantum dots constituted by oxygen vacancies (and clusters thereof). Since for the $V_O$ a charge state transition is intimately coupled to the magnetic properties of the latter, the possibility to study periodically driven transitions in the system also opens up unique avenues to studying the fundamental magnetic phenomena therein. Experimentally it would require operation with a magnetically polarized AFM tip. The presence of STO surface oxygen vacancies and the ease with which they can give and take electrons must play a role in the technologically important interfaces subsequently formed on this substrate.[50,51].

The electronic characterization provided by the AFM mechanical dissipation peaks reported here may be used as an efficient tool for surface analysis, of considerable nanotechnological importance. For instance, since the the in-gap states of a vacancy or vacancy cluster can be seen as a signature or footprint of the latter and the dissipation map, as is discussed in the present study, is directly dependent on the impurity level alignment, one could use pendulum AFM to quickly characterise large areas of a sample with respect to vacancies present thereon.

## Methods

**Tuning fork AFM measurements**. The experiments were performed in two independent UHV chambers. The STM/AFM experiments were performed with a commercial qPlus STM/AFM (Omicron Nanotechnology GmbH) running at low temperature ($T = 5$ K) under UHV and operated by a Nanonis Control System

from SPECS GmbH. We used a commercial tuning-fork sensor in the qPlus configuration (parameters: resonance frequency $f_0 = 23$ kHz, spring constant $k_0 = 1800$ Nm$^{-1}$, quality factor $Q = 13{,}000$ at 5 K). During non-contact friction measurement the oscillation amplitude was always set to 100 pm. The oscillation frequency $f(z)$ and excitation amplitude $A_{exc}$ were recorded simultaneously for different $V$ values applied to the tungsten tip. All STM images were recorded in the constant-current mode. Conductance measurements were performed at constant-height using the lock-in technique with modulation frequency and voltage equal to $f = 653$ Hz and $U = 15$ mV, respectively.

**Pendulum AFM energy dissipation measurements.** The sensitive non-contact friction experiments were performed under UHV by means of a very soft, highly n-doped silicon cantilever (ATEC-Cont from Nanosensors with resistivity $\rho = 0.01$–$0.02$ Ωcm) with spring constant $k = 0.1$ N/m and stiffer, 70 nm gold-coated silicon cantilever (ATEC-NcAu from Nanosensors) with spring constant $k = 50$ N/m. Both were suspended perpendicularly to the surface and operated in the so-called pendulum geometry, meaning that the tip vibrational motion is parallel to the sample surface[36]. Owing to its large stiffness and metallic character, the second tip allowed to perform STM as well as to operate in AFM mode with small oscillation amplitudes allowing for local dissipation measurements. The oscillation amplitudes $A$ of the tip were kept constant by means of a phase-locked loop feedback system at 5 nm for the ultra-sensitive cantilever and at 1 nm, for the stiff sensor. In AFM voltage-dependent measurements the voltage was applied to the sample. The cantilevers were annealed in UHV up to 700 °C for 12 h which results in removal of water layers and weakly bound contaminant molecules from both the cantilever and the tip. After annealing the cantilevers quality factors and the frequencies were equal to $Q = 7 \cdot 10^5$, $f_0 = 11$ kHz and $Q = 6 \cdot 10^4$, $f_0 = 250$ kHz for soft and stiff probes, respectively. It is also known that this long-term annealing leads to negligible amounts of localised charges on the probing tip.

**Non-contact friction coefficient and energy dissipation.** In dissipation measurements the non-contact friction coefficient was calculated according to the standard formula[52]:

$$\Gamma = \Gamma_0 \left( A_{exc}(z)/A_{exc,0} - f(z)/f_0 \right), \qquad (1)$$

where $A_{exc}(z)$ and $f(z)$ are the distance-dependent excitation amplitude and resonance frequency of the cantilever, and the suffix 0 refers to the free cantilever. The distance $z = 0$ corresponds to the point where the tip enters the contact regime, meaning that the cantilever driving signal is saturated and the tunneling current starts to rise. Friction coefficient might be converted into energy dissipation according to the formula:

$$P[\text{eV/cycle}] = \frac{2\pi^2 A^2 f_0}{e} \cdot \Gamma \left[\text{kg s}^{-1}\right], \qquad (2)$$

where $A$ is the oscillation amplitude and $e$ is elementary charge.

**Sample preparation.** The measured sample was 1% Nb-doped SrTiO$_3$ (001) single crystal, which is known to exhibit a variety $[(\sqrt{13} \times \sqrt{13}), (4 \times 2), (4 \times 4), (2 \times 2), (2 \times 1),$ and $(\sqrt{5} \times \sqrt{5})]$ of surface reconstructions[41,42] if annealed in ultra high vacuum. After few cycles of short-term annealing and Ar sputtering the sample was long-term annealed for about 40 min under oxygen-deficient conditions to $T = 850$ °C, $T = 950$ °C, and $T = 1050$ °C, respectively, leading to atomically clean surfaces. The temperature of the sample was controlled with a pyrometer. It is known that the process of UHV, high-temperature annealing leads to sample oxygen deficiency by creation of surface oxygen vacancies[14]. After preparation the samples were transferred into the microscope. The AFM and STM measurements were performed at low temperature $T = 5$ K.

**Data availability.** The data that support the findings of this study are available from the corresponding authors upon reasonable request.

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

## Acknowledgements

O.B. and E.T. gratefully acknowledge the financial support of the ERC Grant No. 320796, MODPHYSFRICT, as well as that of the COST Action MP1303 project. M.K., D.Y., R.P., U.G., and E.M. acknowledge the Swiss National Science Foundation, the Swiss Nanoscience Institute, and the COST Action MP1303 project.

## Author contributions

M.K., E.M., and E.T. conceived the project. M.K., D.Y., R.P., U.G. performed the measurements. M.K. and O.B. analysed experimental data. O.B. and E.T. performed the DFT calculations. O.B., M.K., and E.T. drafted the manuscript, and all authors contributed to the manuscript.

## Additional information

**Competing interests:** The authors declare no competing interests.

