## [Peer Review File · Nature Communications]

Reviewers' comments:

Reviewer #1 (Remarks to the Author):

The manuscript by Kisiel et al. reports on noncontact friction measurements measured on the SrTiO₃ (001) surface. Dissipation is a valuable signal in nc-AFM measurements, which contains information about electron-phonon interactions, charge states of defects, spin flips, etc. However, this signal is rarely interpreted and there is rather little knowledge on how to turn experimental data into quantitative information about the studied material. Data presented in the manuscript are of high quality and the authors are among the top researchers in this field. The topic is novel and possibly of high impact in the fields of nc-AFM and materials science.

Unfortunately, I have to say I see serious flaws in the discussion and data interpretation. I strongly doubt that the dissipation reported in this paper stems from charge and spin transitions of oxygen vacancies, as claimed by the authors. The following three reasons strongly speak against such interpretation:

- 1) All dissipation data in Figure 4 show a perfect symmetry in terms of the applied voltage (positive/negative with respect to the LCPD). However, energy states of oxygen vacancies typically do not possess such symmetry.
- 2) The magnitude of the dissipation seems extremely low to me. Charging and discharging of point defects and the associated dissipation can be observed even in normal cryogenic AFMs (cantilever- or tuning-fork-based). Switching the defect charge state requires considerable band-bending, and the related dissipation is typically at least in order of several meV/cycle.
- 3) Regarding the comparison of soft and stiff cantilevers: The resonance frequencies were 11 and 250 kHz, respectively, and the resulting magnitudes of dissipation are comparable for both cases (in kg/s units). If the dissipation mechanism originated from changing the vacancy charge state, I would expect that a fixed amount of energy is dissipated in each oscillation cycle, which is not the case. Could the authors comment on this?

Regarding the origin of the dissipation, did the authors consider a possible role of electron-phonon interaction? The data in Figure SF6 show that measurements at 5 and 22 K provide qualitatively similar results. Data measured at 90 K show a different type of curve. Finally, the dissipation is gone at 100K. Did the authors consider that SrTiO₃ has bulk phase transitions at

105 and 37 K? (cubic->antiferroelectric->quantum paraelectric).

In principle, the electric field applied to the tip induces lattice distortions (anions and cations become distorted in opposite directions). A similar antiphase motion occurs when an electron becomes confined in the crystal lattice (polaron formation). Could the dissipation peaks arise from some resonance between the field-induced lattice distortions and electron-polarons (donated to the material by oxygen vacancies)?

Besides this, I see several minor problems in the paper

4) Authors assign the reconstruction in Fig. 1d as (2x2), which is incorrect. There are two types of domains with different orientations; this is not possible with a simple (2x2) periodicity. To me the reconstruction looks more like $\sqrt{5} \times \sqrt{5}$ R26.6deg.

5) authors refer to calculations in ref. 13, which were performed on bulk-terminated SrTiO₃. In the experiment, they discuss oxygen vacancies involved in the reconstructed surface. One should at least point out that there is a big difference between energy levels and charge states of these two entities.

6) Figure 1 - please mark crystallographic directions at least in one panel.

7) page 13 - Authors state that the modulation frequency in the lock-in measurements was 653 kHz, which sounds very unlikely. Do they mean 653 Hz?

In conclusion, the topic is important and the data are of high quality, but authors must either provide clear answers to points 1-3, or completely rework the discussion.

Reviewer #2 (Remarks to the Author):

The paper describes transitions in the spectra of charge states trapped by oxygen vacancies on SrTiO₃(100) surface studied by nc-AFM dissipation via pendulum tip. The experimental information is logically built describing first STM characterization of samples annealed at different temperatures and spectroscopies corresponding to electronic states of O vacancies, followed by nc-friction experiments and dissipation spectra maps collected with stiff and soft cantilever tips. Interpretation of the experiments is supported by a simple phenomenological model, presented in Fig. 5, which intuitively explains the experimental results in logical and easy to follow way. Less important experimental data are shown in the Supplementary. I wish to say that, despite the fact that the experimental techniques are not entirely novel, I like the presented results and their interpretation. The manuscript is well written and reasonably easy to follow.

My major problem with this paper is that the paper is pictured at least partly as an experiment-theory work; several theoretical considerations are mentioned in the manuscript, Supplementary Fig.1 shows electronic DOS which is presumably a result of a DFT modeling without mentioning how it was obtained, and also authors' contributions mention DFT calculations, which though, are hard to spot explicitly in the manuscript. My suggestions would be to either strengthen the theory line in the paper by showing more theory results coupled to the experimental results or present the paper as an experimental work only.

Minor points:

- The title mentions spin transitions, and I would like to understand if they can be followed also experimentally and I believe the authors could elaborate more on that point in the text.
- Introduction mentions the very interesting concept of 2DEG of defect states on a surface. However, the present results, at least as I read them, do not show any sign of their presence. I suggest the authors to comment on that.

In summary, this is a nice paper which I do recommend for publication after the above points are sorted out.

Reviewer #3 (Remarks to the Author):

The manuscript "Mechanical dissipation from charge and spin transitions in oxygen deficient SrTiO₃" reports the use of a pendulum atomic force microscopy to measure the charge and spin transitions in oxygen vacancies of STO. My verdict is to consider the manuscript for publication in Nat Comm if the aforementioned major revisions are implemented:

1. Figure 1b: Remove this frame, it is full of feedback ringing and artifacts arising from a broken tip. The rest of the frames of figure 1 are ok, but this frame, to be honest, cannot be published.
2. Nat Comm has broad audience, and the manuscript is written for a specialized journal. I recommend the authors to provide a scheme of how the pendulum AFM works, otherwise, authors that are not familiar with AFMs can follow the research.
3. In the text, it is assumed that the "holes" in the STO surface are due to oxygen vacancies, why? How do you know this is a oxygen vacancy with an AFM?
4. The scale of Figure 4 a is oversaturated at 5×10^{-10} kg/s, can you use different scales for Figure 4 a and b? We are looking the details in frame 4 a. Are there stripes in that are as well?

5. How can you rely in figure 4 b that the CPD moves and it is not an artifact? I suppose this figure is not easy to obtain due to the stability needed, but maybe there is a little bit of drift in the tip height axis?

6. Can you match each or some of the proposed transition of figure5 with the peaks observed in figure 4? I think it would strength the research if yoy can discuss this point.

For the rest of the manuscript, it is well written and the structure is correct.

Referee questions are quoted in light blue italic.

Changes to the manuscript are highlighted in bold red font.

Questions and replies

1. Reviewer #1

The manuscript by Kisiel et al. reports on noncontact friction measurements measured on the SrTiO₃ (001) surface. Dissipation is a valuable signal in nc-AFM measurements, which contains information about electron-phonon interactions, charge states of defects, spin flips, etc. However, this signal is rarely interpreted and there is rather little knowledge on how to turn experimental data into quantitative information about the studied material. Data presented in the manuscript are of high quality and the authors are among the top researchers in this field. The topic is novel and possibly of high impact in the fields of nc-AFM and materials science.

We are delighted with the referee's recognition of the importance and timeliness of the subject we address in our manuscript and for insightful suggestions and comments. Quite generally, it is true that the non-contact dissipation signal is so far difficult to interpret, given the general lack of a quantitatively matching theory. For instance the seminal paper from Stipe, et al. [Phys. Rev. Lett. 87, 096801 (2001)] reported a non-contact dissipation of about 1e-12kg/s while very respectable theory from Volokitin et al. [J. Exp. Theor. Phys. 104, 96 (2007)] points to a much smaller value. We do not claim to provide a resolution to this issue but our study is a further step towards, as the referee puts it, learning to turn experimental data into quantitative information. Well established examples that support our point include Cockins et al. [PNAS 107, 9496 (2010)], and M. Kisiel, et al. [Phys. Rev. Lett. 115, 046101 (2015)]. In our experiment all three different sensors show approximately the same amount of dissipation, an observation which also helps to shed light on the energy dissipation process.

1.1 All dissipation data in Figure 4 show a perfect symmetry in terms of the applied voltage (positive/negative with respect to the LCPD). However, energy states of oxygen vacancies typically do not possess such symmetry.

Our proposed picture is that the tip potential's "shadow" shifts the O-vacancy energy levels. That enables the vertically oscillating AFM to pump electrons between different vacancies. For positive tip bias, electrons are pumped from outside of the shadow to inside, and viceversa for negative bias. Pumping may be enacted via resonant tunneling/thermal hopping as different charge levels inside and outside the shadow coincide and cross as a function of time. A process whose efficiency can indeed be expected to peak at the critical tip distance where, at each tip voltage, the oscillating potential maximum/minimum is exactly reaching the level resonance, because at that point the resonance time is longest. The vacancy – vacancy electron hopping yields a single dissipation peak when the tip potential encloses single vacancy, which is likely the case when the oscillation amplitude is small. The larger size shadow and sweeping range of 5 nm of the pendulum tip can reasonably cover several vacancies, and in that case more dissipation peaks are observed. In both cases the same vacancy levels are involved for charging and discharging in the course of a cycle,

making the to-and-fro dissipation patterns approximately symmetric as proposed and reported. It is hard to specify directly from experiment what energy levels we are precisely involving. Correlating our STS data with DFT calculations is also not as straightforward as it may seem, again difficult because the assumed DFT vacancy structure most likely does not, as noted by the referee, exactly correspond to the actual one because of reconstruction. Yet the relative robustness of DFT results provides at least a strong hint as to the validity of our proposed model. STS reveals a shallow level 0.3 eV below Fermi energy, which according to DFT calculations might be the $q = -2e$ single vacancy charge state. Also in line with Brovko *et. al.* [Phys. Rev. Materials 1, 044405 (2017)] the likeliest low-bias transition for single vacancy could be that between $q = -2e$ and $q = -1e$ charge states.

We add sentences to the manuscript to clarify these issues. We also add the calculated single vacancy level versus local electrochemical potential in Supplementary Materials.

1.2 *The magnitude of the dissipation seems extremely low to me. Charging and discharging of point defects and the associated dissipation can be observed even in normal cryogenic AFMs (cantilever- or tuning-fork-based). Switching the defect charge state requires considerable band-bending, and the related dissipation is typically at least in order of several meV/cycle. Regarding the comparison of soft and stiff cantilevers: The resonance frequencies were 11 and 250 kHz, respectively, and the resulting magnitudes of dissipation are comparable for both cases (in kg/s units). If the dissipation mechanism originated from changing the vacancy charge state, I would expect that a fixed amount of energy is dissipated in each oscillation cycle, which is not the case. Could the authors comment on this?*

Our observed dissipation is indeed of the order of meV/cycle, confirming a correct judgment by the referee, in all cases. Our best dissipation estimated values for soft and stiff cantilevers, and for the tuning fork, are very close, 12 meV/cycle, 18 meV/cycle and 14 meV/cycle, respectively. We used the following conversion formula: P [eV/cycle] = $(2 \pi^2 A^2 f_0 / e) \cdot \Gamma$ [kg/s], where A is the oscillation amplitude and f_0 is the oscillation frequency. The band bending need not be large, as it is probably mitigated by vacancy-related surface states that pin the Fermi level. A large concentration of surface O vacancies pins the Fermi level close to the conduction band bottom at the STO surface. Given that starting point, small local changes of the chemical potential induced by the oscillating can be effective in switching charge states giving rise the observed dissipation.

We have changed the manuscript to comment on dissipation value given by different sensors and compare it to other measurements.

1.3 *Regarding the origin of the dissipation, did the authors consider a possible role of electron-phonon interaction? The data in Figure SF6 show that measurements at 5 and 22 K provide qualitatively similar results. Data measured at 90 K show a different type of curve. Finally, the dissipation is gone at 100K. Did the authors consider that SrTiO3 has bulk phase transitions at 105 and 37 K? (cubic->antiferroelectric->quantum paraelectric).*

The role of electron-phonon finally, as we see it, is certainly important in determining the exact vacancy level positions. That aspect is automatically taken into account in the DFT calculations, where the atomic relaxation pattern near the vacancy is crucially dependent upon the charge state.

Concerning temperature dependence we should have mentioned, actually, that each temperature demands a new separate measurement, and the (x,y) tip location changes each time. However, the detailed dissipation peak patterns depend on the tip location, reflecting the uneven vacancy distribution. Because of that, a meaningful study of individual dissipation peaks as a function of temperature cannot be done.

Next, we are aware of the antiferrodistortive $T_C \sim 105$ K transition and of the 37 K quantum paraelectric onset of SrTiO₃, subjects of previous work by some of us, (M. Kisiel, *et al.* [Phys. Rev. Lett. 115, 046101 (2015)], K.A. Mueller *et al.* [Z. Phys. 84, 277583 (1991)]). In the 2015 AFM work in particular, it was found that pendulum AFM dissipation occurred by coupling to critical strain fluctuations very close to $T = T_C$, a different mechanism and unrelated to O vacancies.

Present data show that our novel dissipation peaks, only seen with large surface vacancy concentrations and attributed to charging and discharging of O vacancies, persist up to temperature as high as $T = 90$ K without any clear connections with these transitions and order parameters. Our best explanation is that, as in other quantum dots, dissipation peaks are only visible so long as the charging energy involved is larger than the thermal energy $k_B T$. That suggests that the “shadow potential” projected by the tip and the consequent changes of vacancy states only work as proposed within an energy scale of order $k_B T \sim 10$ meV. This is not unreasonable, since the vacancy “redox” levels, all lie very close to the STO conduction band and can possibly be switched by a ~ 10 meV tip-induced shift.

We are now mentioning this aspect in the revised manuscript.

- 1.4 *In principle, the electric field applied to the tip induces lattice distortions (anions and cations become distorted in opposite directions). A similar antiphase motion occurs when an electron becomes confined in the crystal lattice (polaron formation). Could the dissipation peaks arise from some resonance between the field-induced lattice distortions and electron-polarons (donated to the material by oxygen vacancies)?*

This interesting question concerns the detailed coupling mechanism through which the tip manages to shift the electron chemical potential at the vacancy site. Indeed, such coupling can have one origin that involves a tip-induced local atomic distortion, (designated “polaronic” by the referee), plus another purely electronic origin, a Coulomb or van der Waals shift of the vacancy level, that would work even if the atomic structure was rigid under the tip. As always in AFM, we do not have easy access to ways of distinguishing between the two. Tentatively, we think that the latter, purely electronic coupling, should be dominant in this case because the atomic distortion would be asymmetric between positive and negative tip voltage, whereas the observed dissipation peak pattern is reasonably symmetric. It should be noted in any case that the proposed dissipation mechanism works irrespective of the precise nature of the coupling.

Besides this, I see several minor problems in the paper:

- 1.5 *Authors assign the reconstruction in Fig. 1d as (2×2) , which is incorrect. There are two types of domains with different orientations; this is not possible with a simple (2×2) periodicity. To me the reconstruction looks more like $\sqrt{5} \times \sqrt{5}$ R26.6 deg.*

We agree with referee, and we are grateful for the opportunity to correct this slip. Our STM data are basically identical to the $\sqrt{5} \times \sqrt{5}$ surface reconstruction by Tanaka *et al.* [Japanese Journal of Applied Physics 32, 1405 (1993)], with domains rotated by

approximately 26 deg from the <100> bulk direction.

We made the changes in the manuscript and the Figure 1.

- 1.6 *Authors refer to calculations in ref. 13, which were performed on bulk-terminated SrTiO₃. In the experiment, they discuss oxygen vacancies involved in the reconstructed surface. One should at least point out that there is a big difference between energy levels and charge states of these two entities.*

We gladly concur with this observation. DFT calculations were not easy, because of several factors. Lack of structural data, first of all, made every starting geometry a dive in the dark. Very large size cells were needed in order to include enough STO “bulk”, and in order to keep vacancies reasonably separated from each other. Finally, very lengthy relaxations were called for in every case and of course in every geometry and state of charge. Yet, following up and refining calculations of Ref. 13 for the unreconstructed surface, we did calculations for the vacancy levels for three different flavors of the vacancy on the simplest case of 2 × 2 reconstruction.

A figure describing these three flavors is now added to the supplementary material.

In the 2 × 2 reconstruction case we could still manage the extra-large unit cell, although with the additional assumption of an unreconstructed second layer. With all the caveats implied by these assumptions, we eventually found that the results were different from each other and from the unreconstructed surface, although not wildly so. Based on that qualified uncertainty, and on the fact that a similar calculation for the $\sqrt{5} \times \sqrt{5}$ was anyway simply out of reach, it seemed in the end more reasonable to use the unreconstructed surface vacancy levels, theoretically more robust, as our semi-quantitative guidance in the interpretation of the experimental results.

- 1.7 *Figure 1 - please mark crystallographic directions at least in one panel.*

The crystallographic directions are marked in a new Fig. 1(c).

- 1.8 *page 13 - Authors state that the modulation frequency in the lock-in measurements was 653 kHz, which sounds very unlikely. Dis they mean 653 Hz?*

That is a typo. It is, of course, 653 Hz. Thank you for pointing it out.

In conclusion, the topic is important and the data are of high quality, but authors must either provide clear answers to points 1-3, or completely rework the discussion.

Above we have elaborated in depth on the three principal question raised by the referee.

2. Reviewer #2

- 2.1 *My major problem with this paper is that the paper is pictured at least partly as an experiment-theory work; several theoretical considerations are mentioned in the manuscript, Supplementary Fig.1 shows electronic DOS which is presumably a result of a DFT modeling without mentioning how it was obtained, and also authors'*

contributions mention DFT calculations, which though, are hard to spot explicitly in the manuscript. My suggestions would be to either strengthen the theory line in the paper by showing more theory results coupled to the experimental results or present the paper as an experimental work only.

The referee's suggestion to strengthen the theory part of the manuscript is quite reasonable, and we are pleased to follow it. The amount of calculations we did was in fact not small, as we now describe in Supplementary Material. As a result of that however, as detailed in our response to Referee 1, the overall results were at the same time generic and quantitatively uncertain, suggesting that we use after all the simpler unreconstructed surface vacancy levels as our main qualitative theoretical guidance.

In that optics, we now add as a useful theory graph the unreconstructed vacancy energy level for $q = -e$ and $q = -2e$ versus chemical potential, the quantity that in our model is modulated in time by the oscillating tip. On the graph the crossing of the energy for different charge states results in dissipation peak.

Minor points:

- 2.2. *The title mentions spin transitions, and I would like to understand if they can be followed also experimentally and I believe the authors could elaborate more on that point in the text.*

The isolated O vacancy on the surface of a band insulator will, just like an atom or a quantum dot, possess a generally nonzero quantized spin, at least for odd charging. In our case the $q = -e$ state should have $S = 1/2$, while the $q = -2e$ drops to $S = 0$ due to the antiferromagnetic coupling of the two dangling Ti ions. Experimentally it is not possible to detect spin with a non-magnetic AFM tip, but its presence/absence on either sides of a dissipation peak is an interesting information provided by our theoretical calculations. If, as a result of a large vacancy concentration the STO surface should develop a 2D Fermi sea, ferromagnetically polarized, as advocated by some authors [Z. Q. Liu, et.al., Phys. Rev. B **87**, 220405(R), (2013)] that might influence and change our level picture.

We have added additional sentence into the manuscript to comment on that.

- 2.3 *Introduction mentions the very interesting concept of 2DEG of defect states on a surface. However, the present results, at least as I read them, do not show any sign of their presence. I suggest the authors to comment on that.*

As described, e.g., by A. F. Santander-Syro *et al.*, Nature 469, 189–193 (2011) – the 2D electron gas does form on a the heavily vacancy-doped STO surface. We have no direct evidence of its presence in our case, but we cannot rule it out either. If present, it would provide an electron reservoir for the transition between different states of charge of the vacancy.

This is now marked in Figure 5.

In summary, this is a nice paper which I do recommend for publication after the above points are sorted out.

We did our best.

3. Reviewer #3

The manuscript “Mechanical dissipation from charge and spin transitions in oxygen deficient SrTiO₃” reports the use of a pendulum atomic force microscopy to measure the charge and spin transitions in oxygen vacancies of STO. My verdict is to consider the manuscript for publication in Nat Comm if the afore-mention major revisions are implemented.

- 3.1 *Figure 1b: Remove this frame, it is full of feedback ringing and artifacts arising from a borken tip. The rest of the frames of figure 1 are ok, but this frame, to be honest, cannot be published.*

Agreed. We removed Figure 1(b).

- 3.2 *Nat Comm has broad audience, and the manuscript is written for a specialized journal. I recommend the authors to provide a scheme of how the pendulum AFM works, otherwise, authors that are not familiar with AFMs can follow the research.*

Agreed.

We added an illustration of pendulum AFM to the Supplementary.

- 3.3 *In the text, it is assumed that the “holes” in the STO surface are due to oxygen vacancies, why? How do you know this is a oxygen vacancy with an AFM?*

Fig. 1 presents STM data at $U_{\text{tip}} = -1\text{V}$, and not AFM. Our statement in Fig. 1(d) caption that dark spots are O vacancies stems from the simple consideration that there would be a large density of STO empty states if at these spots the O atom was present, as is seen everywhere except at the dark spots. Moreover, our STS data (Fig. 3) reproduce a similar electronic structure as that previously reported [H. Tanaka et al., Japanese Journal of Applied Physics 32, 1405 (1993)] for oxygen reduced sample.

We added this explanation in the revised text.

- 3.4 *The scale of Figure 4 a is oversaturated at 5×10^{-10} kg/s, can you use different scales for Figure 4 a and b? We are losing the details in frame 4 a. Are there stripes in that are as well?*

This choice was made because there was nothing interesting to see at these large peaks. We chose this scale in order to emphasize the fine features.

Additional figures are now added to the Supplementary.

Dissipation map taken with local probe with a stiffness $k = 50\text{N/m}$, while forward and backward voltage sweep.

Plotted differently, the same dissipation signal versus tip-sample distance and for constant tip-sample voltage $V = 3\text{V}$, clearly demonstrates the dissipation saturation as the tip approaches the surface.

Distance dependent dissipation measured with cantilever probe ($k = 50\text{N/m}$) oscillating with amplitude $A = 1\text{nm}$. Tip-sample voltage is constant and equal to $U = 3\text{V}$.

3.5 *How can you rely in figure 4 b that the CPD moves and it is not an artifact? I suppose this figure is not easy to obtain due to the stability needed, but maybe there is a little bit of drift in the tip height axis?*

The figure is obtained in a commonly accepted regular force spectroscopy mode, and is actually less complicated than reported [Mehmet Z. Baykara and Udo D. Schwarz (3D Force Field Spectroscopy in Noncontact Atomic Force Microscopy), S. Morita *et al.* (Springer International Publishing Switzerland 2015, DOI 10.1007/978-3-319-15588-3_2)] grid spectroscopy force field since it is a measurement on a single spot. Concerning stability the measurements were performed at $T = 5\text{K}$, therefore the z -axis drift is negligible in a distance range of several nm.

Moreover, we acquire data while voltage sweeping forward and backward (in the

manuscript we showed only forward), precisely as a cross-check against artifacts. Both forward and backward sweeps showed similar behaviour and CPD change. Finally, the effect is fully reproducible. Below the different set of data is shown for the measurement performed at $T = 5\text{K}$ and on different surface spot.

Dissipation maps taken on reduced STO surface with pendulum geometry at oscillation amplitude 5nm. Changes of CPD value while switching between different dissipation branches are reproducible and consistent with charge accumulation below the tip.

To shed more light let us compare the dissipation maps obtained on NbSe_2 sample (M. Langer *et al.* [Nature Materials 13, 173177 (2014)]). In case of NbSe_2 the observed giant dissipation was a result of hysteretic switching of Charge Density Wave, being in fact purely periodic lattice deformation. The dissipation mechanism is therefore fully phononic and no significant charge accumulation is expected below the oscillating tip. There are three dissipation branches and no CPD change is observed while switching from one to another branch. There are completely symmetric with respect to CPD.

CDW (phononic) dissipation map for NbSe_2 sample shows three symmetric dissipation peaks and no CPD change while switching

between different dissipation branches.

- 3.6 *Can you match each or some of the proposed transition of figure5 with the peaks observed in figure 4? I think it would strength the research if you can discuss this point.*

Unfortunately, we cannot specify exactly from theory or from experiment what levels we are precisely observing. We can speculate by comparison of STS data with DFT calculations. STS data reveal shallow energy state localized 0.3 eV below Fermi energy, which according to DFT calculations could be the $q = -2e$ single vacancy charge state. The energy formation graph now added to Supplementary suggests that the observed dissipation transitions between $q = -2e$ and $q = -1e$ single vacancy states.

We mention this in a revised version of the manuscript. Concerning Figure 5 we describe different vacancy charging scenario I-IX (electron exchange between neighbouring vacancies, vacancy and STO conduction band and vacancy and 2DEG known to exists on reduced STO surface) in the Figure caption.

Reviewers' comments:

Reviewer #1 (Remarks to the Author):

The authors have carefully considered my comments and now I recommend the paper for publication in Nature Communications. After reading the rebuttal it is more clear to me what kind of dissipation has been observed and what the authors want to state in the paper. I would suggest a minor revision; I have a few comments on how the work is presented.

1) The paper indicates many possibilities for the origin of the dissipation and the main storyline seems to prefer switching the charge state of single oxygen vacancies. This scenario is strongly preferred in the abstract and introduction. Here the reader forms his/her opinion and it is difficult to change it later.

In my opinion, the single-vacancy mechanism is very unlikely. Switching the charge state of a point defect is well known from STM experiments (see for example [PRL 94, 076801 (2005)] or [PRL 101, 076103 (2008)] or ref. 39:Proc.Natl. Acad. Sci. USA 107 (21) (2010) 9496-9501). Tip-induced charging of point defects occurs at relatively close tip-sample distances and is well localized around the defect. The corresponding dissipation signal appearing in AFM is as a ring of enhanced dissipation centered on the defect. This is clearly inconsistent with the presented data.

2) I find three key experimental findings in the manuscript: First, the dissipation appears already at long tip-sample distances (even above 10 nm). Second, the dissipation disappears above 80-90 K. Third, the crystal reduction state is important. I would mention these findings already at the beginning of the text (abstract or introduction).

3) In the discussion, authors mention that the dissipation could originate from a collective effect involving several electrons and oxygen vacancies. My feeling is that this is more likely than charging a single point defect. The dissipation signal seems to disappear rather abruptly at certain temperature, which indicates that some sort of collective behavior is involved.

4) In Figs. 2 and 4, the units of [eV/cycle] should be included together with kg/s. Nat. Communications is a journal addressed to a broad audience and the used units may be difficult to imagine for many scientists. I would also mention the conversion formula somewhere.

Reviewer #2 (Remarks to the Author):

The authors have given a very comprehensive response to all three reviewers and modified the

manuscript accordingly. I was mainly concerned about the role of theory in the paper. I understand now the situation and consider that point sorted out. I believe the improved paper can be published as is.

Reviewer #3 (Remarks to the Author):

Thanks for answering and considering my comments.

I am happy with the answers given and I can recommend now the manuscript for publication

Responses to Reviewers' Comments:

Referee questions are quoted in light blue italic.

Questions and replies

1. 1. Reviewer #1

The authors have carefully considered my comments and now I recommend the paper for publication in Nature Communications. After reading the rebuttal it is more clear to me what kind of dissipation has been observed and what the authors want to state in the paper. I would suggest a minor revision; I have a few comments on how the work is presented.

1. 1.1 *The paper indicates many possibilities for the origin of the dissipation and the main storyline seems to prefer switching the charge state of single oxygen vacancies. This scenario is strongly preferred in the abstract and introduction. Here the reader forms his/her opinion and it is difficult to change it later.*

In my opinion, the single-vacancy mechanism is very unlikely. Switching the charge state of a point defect is well known from STM experiments (see for example [PRL 94, 076801 (2005)] or [PRL 101, 076103 (2008)] or ref. 39:Proc.Natl. Acad. Sci. USA 107 (21) (2010) 9496-9501). Tip-induced charging of point defects occurs at relatively close tip-sample distances and is well localized around the defect. The corresponding dissipation signal appearing in AFM is as a ring of enhanced dissipation centered on the defect. This is clearly inconsistent with the presented data.

We adhere to this suggestion, and made changes in the abstract, in the introduction, and in the text. There, we indicate that while a collective mechanism involving more the multi- vacancy system is the likeliest occurrence, but we suggest at the same time

that the single vacancy can be effectively used as a model to guide our understanding.

1. 1.2 *I find three key experimental findings in the manuscript: First, the dissipation appears already at long tip-sample distances (even above 10 nm). Second, the dissipation disappears above 80-90 K. Third, the crystal reduction state is important. I would mention these findings already at the beginning of the text (abstract or introduction).*

These elements are now specified at the end of the introduction.

1. 1.3 *In the discussion, authors mention that the dissipation could originate from a collective effect involving several electrons and oxygen vacancies. My feeling is that this is more likely than charging a single point defect. The dissipation signal seems to disappear rather abruptly at certain temperature, which indicates that some sort of collective behavior is involved.*

Agreed.

1. 1.4 *In Figs. 2 and 4, the units of [eV/cycle] should be included together with kg/s. Nat. Communications is a journal addressed to a broad audience and the used units may be difficult to imagine for many scientists. I would also mention the conversion formula somewhere.*

Done. The conversion formula is specified in the Methods section.

1. 2. Reviewer #2

The authors have given a very comprehensive response to all three reviewers and modified the manuscript accordingly. I was mainly concerned about the role of theory in the paper. I understand now the situation and consider that point sorted out. I believe the improved paper can be published as is.

Thank you.

1. 3. Reviewer #3

Thanks for answering and considering my comments.

I am happy with the answers given and I can recommend now the manuscript for publication!

Thank you.